TECHNICAL RELEASE

# ensemblQueryR: fast, flexible and high-throughput querying of Ensembl LD API endpoints in R

Aine Fairbrother-Browne[1,2,3,*], Sonia García-Ruiz[1,4],
Regina Hertfelder Reynolds[1,4], Mina Ryten[1,4,†] and Alan Hodgkinson[2,†]

1 Department of Genetics and Genomic Medicine Research & Teaching, UCL GOS Institute of Child Health, London, UK
2 Department of Medical and Molecular Genetics, School of Basic and Medical Biosciences, King's College London, London, UK
3 Department of Neurodegenerative Disease, Queen Square Institute of Neurology, UCL, London, UK
4 NIHR Great Ormond Street Hospital Biomedical Research Centre, University College London, London, UK

## ABSTRACT

We present ensemblQueryR, an R package for querying Ensembl linkage disequilibrium (LD) endpoints. This package is flexible, fast and user-friendly, and optimised for high-throughput querying. ensemblQueryR uses functions that are intuitive and amenable to custom code integration, familiar R object types as inputs and outputs as well as providing parallelisation functionality. For each Ensembl LD endpoint, ensemblQueryR provides two functions, permitting both single- and multi-query modes of operation. The multi-query functions are optimised for large query sizes and provide optional parallelisation to leverage available computational resources and minimise processing time. We demonstrate improved computational performance of ensemblQueryR over an exisiting tool in terms of random access memory (RAM) usage and speed, delivering a 10-fold speed increase whilst using a third of the RAM. Finally, ensemblQueryR is near-agnostic to operating system and computational architecture through Docker and singularity images, making this tool widely accessible to the scientific community.

**Submitted:** 23 May 2023

**Subjects** Software and Workflows, Bioinformatics, Genetics, Software Engineering

\* Corresponding author. E-mail:
ucbtas8@ucl.ac.uk

† Contributed equally.

Preprint submitted at https://doi.org/10.48550/arXiv.2308.06792

## STATEMENT OF NEED

### Background

Linkage disequilibrium (LD) is the non-random association of alleles arising from different loci [1]. In population genetics, LD is a measure of the frequency with which an allele of one variant is correlated with an allele of a proximal variant within a particular population [2]. There are many applications for LD measures in genomics workflows. For example, in the context of genome-wide association studies (GWAS), which have been used to detect associations between genetic variants and a wide range of human phenotypes, downstream interrogation of local LD structure is required to identify the potential 'causal' variant at a nominated locus that exerts an effect on the downstream phenotype. Equally, in expression quantitative trait loci (eQTL) analyses, which aim to uncover associations between genetic variants and the expression of a *cis* or *trans* gene (eGene), LD information is required for the identification of the potential causal variant affecting the expression of

the eGene. Further downstream, LD information is useful for functional annotations, where genetic variants or regions in LD with a target variant can aid in the identification of biological processes that might be affected by the GWAS- or eQTL-implicated target variant. As such, it is important that the LD information for a range of human populations can be easily queried by researchers in an efficient and accessible way.

Despite the widespread usage of LD measures in genomic research, the majority of tools available at present are web-based. Although these offer user-friendly interfaces and can be useful for one-off or small queries, they do not promote reproducibility and are not suited to workflow-oriented researchers wishing to submit multiple large queries. Programmatic tools offer a solution to these problems; however, very few tools for the retrieval of LD metrics exist.

To our knowledge, only one R package provides a programmatic interface for LD metric retrieval. LDlinkR (version 1.2.3) [3] provides an R-based interface to the web-based tool LDlink [4], permitting retrieval of LD metrics using a range of query types. However, LDlinkR has a number of key limitations with respect to speed and query handling. Firstly, the user is required to obtain an access token by signing up on the NCBI website, which is then supplied as an argument to all LDlinkR functions. This requirement is in place to limit user queries, meaning that attempts to speed up the tool using parallelisation easily exceed query limits and cause the tool to return timeout errors. This can result in the user's access token being blocked. Secondly, a number of functions for retrieving LD metrics are configured for singular queries only –such as the LDpair and LDproxy functions –meaning that the user must write custom code to submit more

than one query at one time. As such, although LDlinkR is a useful programmatic alternative to the LDlink web tool, it is not suited to fast, high-throughput multi-query retrieval of LD metrics.

Ensembl (RRID:SCR_002344) is another widely used source of LD metrics, offering an application programming interface (API) that supports an array of query configurations [5, 6]. However, some challenges are presented by direct API usage as its usage requires some technical expertise. Additionally, it is not easily integrable with typical R workflows, precludes the input of standard R objects (such as data frames, lists or vectors), does not output data in an intuitive format and is not easily adaptable to high-throughput workflows. To our knowledge, no R package has been developed to facilitate querying the Ensembl API and, in particular, to retrieve Ensembl LD metrics. In light of this, and to address the limitations of current tools, we present ensemblQueryR. Our R package provides fast, efficient, user-friendly querying of Ensembl LD data, with a focus on intuitive, high-throughput R workflow integration.

ensemblQueryR has been made freely available (DOI: 10.5281/zenodo.7837882) [7, 8]. The package can also be used in Docker (RRID:SCR_016445) [9] or Singularity [10] containers, for which the images can be found on Docker Hub [11] or the Singularity image repository [12].

## Implementation
### *Our approach*
ensemblQueryR provides a suite of functions that wrap around three Ensembl API 'endpoints'. These endpoints operate to retrieve data from Ensembl databases through the following query configurations:



**Table 1.** The functions comprising the ensemblQueryR package and their relationship to the three LD Ensembl API endpoints.

| ensemblQueryR function | Ensembl API endpoint | Arguments | Output | Description |
|---|---|---|---|---|
| ensemblQueryGetPops | Information [5] | N/A | A list of human populations for which LD metrics can be retrieved. | This function retrieves a list of the Ensembl populations for which LD metrics can be queried (for further information on the available populations, see Ensembl [13]). These can be supplied to the 'pop' argument across ensemblQueryR functions. |
| pingEnsembl | Information [5] | N/A | An integer (and a message) to indicate the status of the Ensembl API. Returns 1 and reports "Server OK." if the server is up. | This function checks and informs the user of the status of the Ensembl API. |
| ensemblQueryLDwithSNPwindow | Window [5] | rsid<br>r2<br>d.prime<br>window.size<br>pop | A data frame with five columns: 'query' (the variant input to 'rsid'), snp_in_ld (variant(s) in LD with 'query'), r2 (r-squared statistic), d_prime (D' statistic), population_name (the population supplied to 'pop'). | This function retrieves variants in LD with the query variant within a given genomic window. |
| ensemblQueryLDwithSNPwindowDataframe | Window [5] | in.table<br>r2<br>d.prime<br>window.size<br>pop<br>cores | A data frame with five columns: 'query' (the variant input to 'rsid'), snp_in_ld (variant(s) in LD with 'query'), r2 (r-squared statistic), d_prime (D' statistic), population_name (the population supplied to 'pop'). | This function takes a data frame with a column of variant rsIDs (Reference SNP cluster IDs). It retrieves variants in LD with each query variant within a given genomic window. |
| ensemblQueryLDwithSNPpair | Pair [5] | rsid1<br>rsid2<br>pop | A data frame with five columns: 'query1' (the variant input to 'rsid1'), 'query2' (the variant input to 'rsid2'), r2 (r-squared statistic), d_prime (D' statistic), population_name (the population supplied to 'pop'). | This function takes a pair of rsIDs and retrieves their LD metrics (D' and $R^2$). |
| ensemblQueryLDwithSNPpairDataframe | Pair [5] | in.table<br>pop<br>cores | A data frame with five columns: 'query1' (the variant input to 'rsid1'), 'query2' (the variant input to 'rsid2'), r2 (r-squared statistic), d_prime (D' statistic), population_name (the population supplied to 'pop'). | This function takes a data.frame containing paired rsIDs, retrieving LD metrics (D' and $R^2$) for all pairs. |
| ensemblQueryLDwithSNPregion | Region [5] | chr<br>start<br>end<br>pop | A data frame with eight columns: 'query_chr' (the query chromosome supplied to 'chr'), 'query_start' (the query start coordinate supplied to 'start'), 'query_end' (the query end coordinate supplied to 'end'), 'rsid1' (variant one of two in the pair), 'rsid2' (variant two of two in the pair), r2 (r-squared statistic), d_prime (D' statistic), population_name (the population supplied to 'pop'). | This function takes a genomic coordinate, retrieving LD metrics (D' and $R^2$) for all rsID within the defined region. |
| ensemblQueryLDwithSNPregionDataframe | Region [5] | in.table<br>pop<br>cores | A data frame with eight columns: 'query_chr' (the query chromosome supplied to 'chr'), 'query_start' (the query start coordinate supplied to 'start'), 'query_end' (the query end coordinate supplied to 'end'), 'rsid1' (variant one of two in the pair), 'rsid2' (variant two of two in the pair), r2 (r-squared statistic), d_prime (D' statistic), population_name (the population supplied to 'pop'). | This function takes a data frame containing genomic coordinate(s) and retrieves LD metrics (D' and $R^2$) for all rsID within the defined region(s). |

1. Window: retrieval of the LD metrics for a variant and all the other variants in a window around the target variant;
2. Pair: retrieval of the LD metrics between a pair of target variants;
3. Region: retrieval of the LD metrics between all pairs of variants in a defined target region.

Single-query and multi-query wrapper functions are provided for each of these Ensembl API endpoints, all of which are described in detail in Table 1.

To make ensemblQueryR useful in a high-throughput context, the main challenge is that the Ensembl API endpoints are configured to handle single queries. To address this, ensemblQueryR's three multi-query functions (with names ending in 'Dataframe', as described in Table 1) take data frame objects as input, where each row needs to be

submitted as a separate query to the Ensembl API. The base R lapply function (Version 4.0.5) [14] is then used to apply the corresponding single-query function over the input data frame, iteratively formulating an API query from each data frame row.

Building on ensemblQueryR's high-throughput capabilities, we implemented optional parallelisation for all multi-query functions. Each multi-query function (those with names ending in 'Dataframe' in Table 1) has an argument that allows the user to set a number of 'cores' to parallelise the query across. Using this functionality can significantly reduce run-time, particularly for larger queries where the parallelisation overheads represent a small proportion of the overall memory requirements. For example, with a query size of 1,000, the ensemblQueryLDwithSNPpairDataframe function running on a single core takes ~2.4 min, whereas ten cores speed the process up by 22 times, reducing the execution time to ~0.11 min (System tested on: Ubuntu server 16.04 LTS with kernel version 4.4.0-210-generic, total RAM 251G).

Consistency of the data output format is an important feature of ensemblQueryR, making it amenable to R workflow integration. All functions (single- and multi-query) return a data frame object, including instances where the query returns a null result or an error. This consistent output format simplifies writing custom code, allowing users to incorporate ensemblQueryR functionality into bespoke workflows. It is important to note that when data cannot be retrieved, for example, if a variant is not found in Ensembl databases or the user input is invalid, console messages alert the user of this and a data frame row containing 'NA' values is returned for that query.

### Benchmarking

LDlinkR is an alternative R package that offers LD metric retrieval. As such, it was important to benchmark against this tool to demonstrate the utility of ensemblQueryR for high-throughput querying. Of the functions contained in the LDlinkR and ensemblQueryR packages, two functions are particularly comparable in their functionality. Both LDpair (from LDlinkR) and ensemblQueryLDwithSNPpair (from ensemblQueryR) take a pair of reference SNP cluster identifiers (rsIDs) as input, while the output is a table containing the LD metrics for the query pair. As such, these functions were selected for benchmarking. To compare the performance of the two functions, the computation speed and RAM usage at three query sizes representing a range of throughputs –100, 1,000 and 10,000 queries –were assessed (Figure 1). For each function and query size combination, performance (speed and peak RAM usage) was tested ten times to account for temporal fluctuations in processing speed and peak RAM usage, thus enabling a precise performance assessment.

Firstly, comparing execution speed, we found that, on average (across the ten tests), ensemblQueryLDwithSNPpair was 10.2 times faster in the 100-query test, taking an average of 0.208 min compared to the 2.12 min for LDpair (Figure 1b). The 1,000-query test found that ensemblQueryLDwithSNPpair was, on average, 9.92 times faster than LDlinkR, taking an average of 1.97 min compared to the 19.5 min for LDpair. Finally, in the 10,000-query test, LDpair was unable to produce a final results table in seven out of ten tests, in these instances returning an error message ('Bad Gateway (HTTP 502)'). In contrast, ensemblQueryLDwithSNPpair produced a final results table in all tests, demonstrating its reliability for high-throughput querying. Looking at the only three successful tests of

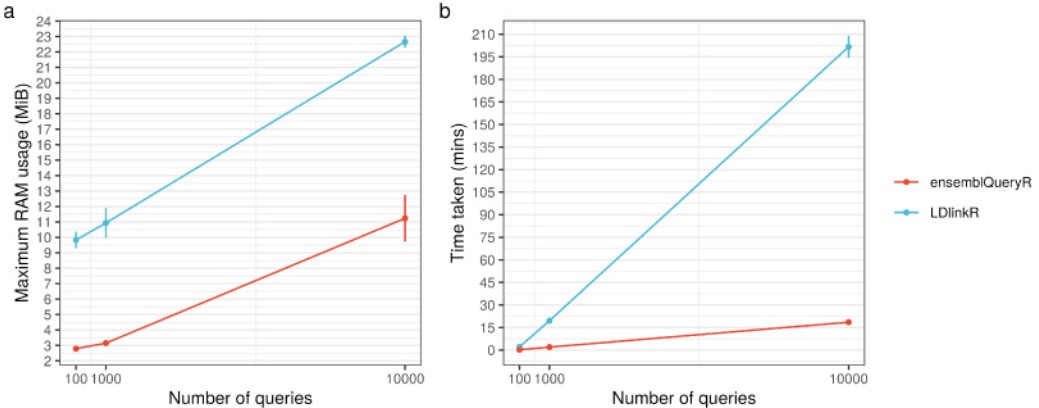

**Figure 1.** Comparison of performance metrics between analogous functions of the ensemblQueryR and LDlinkR R packages. (a) Plot showing, for query sizes of 100, 1,000 and 10,000, the maximum RAM usage (mebibytes, MiB) during the execution of ensemblQueryR's ensemblQueryLDwithSNPpair (red) and LDlinkR's LDpair (blue). (b) Plot showing, for query sizes of 100, 1,000 and 10,000, the execution time (minutes, mins) of ensemblQueryR's ensemblQueryLDwithSNPpair (red) and LDlinkR's LDpair (blue).

LDpair, we found that ensemblQueryLDwithSNPpair was, on average, 10.9 times faster than LDpair, taking an average of 18.5 min compared to the 202 min (>3 h) for LDpair. These speed improvements are likely due to request rate limits from the server side, which are higher for Ensembl, thus enabling fast concurrent or parallel requests.

Secondly, we compared the peak RAM usage –the maximum RAM utilised at any time during function execution –between ensemblQueryLDwithSNPpair and LDpair (Figure 1a). We found that across query sizes, ensemblQueryLDwithSNPpair had approximately a third (range: 20.8–49.6%) of the peak RAM usage of LDpair. These peak RAM usage improvements are likely due to a focus on within-function reductions of intermediate object storage and a reduction of the number of operations carried out within the ensemblQueryLDwithSNPpair function.

We conclude that, by comparing analogous functions (ensemblQueryLDwithSNPpair and LDpair), ensemblQueryR provides a performance improvement over LDlinkR with respect to both speed (×10) and memory usage (1/3), underscoring the utility of our tool in the context of high-throughput workflows.

### Usage
The following code provides an implementation example for ensemblQueryR. In this case, the target variant was taken from the OpenTargets homepage [15] with rsID rs4129267. To find out which variants, within a set of genomic window sizes, are in LD ($R^2 > 0.8$ and $D' > 0.8$) with the target variant, the ensemblQueryLDwithSNPWindow function was implemented as in Figure 2.

### Methods
*Docker and singularity images.* In order to make this tool widely accessible to the research community, we have provided a Docker image (Figure 3). It is based on a Rocker (RRID:SCR_024215) [16] with R version 4.0.0 and Ubuntu version 20.04 LTS (Focal Fossa) pre-installed. To start a container using the 'ensemblqueryr' Docker image and launch an



```
# load package
library(ensemblQueryR)

# get list of available populations for which LD metrics can be
returned
ensemblQueryR::ensemblQueryGetPops()

# implement ensemblQueryLDwithSNPwindow to get variants in a window of
size 500kb centered on rs4129267, setting r2 and d.prime to constrain
the variants returned
ensemblQueryR::ensemblQueryLDwithSNPwindow(rsid="rs4129267",
  r2=0.8,
  d.prime=0.8,
  window.size=500,
  pop="1000GENOMES:phase_3:EUR")
```

**Figure 2.** Working example of an ensemblQueryR implementation.

interactive R session within which the tool can be used, the following command can be executed on the command line:

```
docker pull ainefairbrotherbrowne/ensemblqueryr:1.0; \
docker run -t -d --name ensemblqueryr
ainefairbrotherbrowne/ensemblqueryr:1.0; \
docker exec -i -t ensemblqueryr R
```

To mount a local volume that contains a file of variant IDs, the following command can be used:

```
docker pull ainefairbrotherbrowne/ensemblqueryr:1.0; \
docker run -t -d --name ensemblqueryr ainefairbrotherbrowne/ensemblqueryr:1.0
-- path/to/volume; \
docker exec -i -t ensemblqueryr R
```

Additionally, for use-cases wherein the user does not have sufficient permissions to use Docker, such as on HPC clusters, a singularity image can be used instead. To generate a singularity image, we converted the ensemblQueryR docker image [11] into a singularity image using the Docker-based tool docker2singularity (Version 2.6) [17]. This image can be found on the sylabs.io singularity repository [12]. The singularity image can be pulled and executed, launching a command line R session as follows:

```
singularity pull --arch amd64
library://ainefairbrother/ensemblqueryr/ensemblqueryr:sha256.e387ea11ae4eaea8
f94d81c625c2c1d5a22dd351858ebcd03910a7736d76ca30; \
singularity exec
ensemblqueryr_sha256.e387ea11ae4eaea8f94d81c625c2c1d5a22dd351858ebcd03910a773
6d76ca30.sif R
```

*Benchmarking.* To benchmark our package against LDlinkR, we first selected functions that performed similar tasks. The most comparable functions in this context were LDlinkR's

```
# use rocker 4.0.0 and Ubuntu 20, as these resemble closely the development OS for ensemblQueryR
FROM rocker/r-ver:4.0.0

# system libraries of general use - informed by https://tinyurl.com/ypnkbrpb
RUN apt-get update && apt-get install -y \
    sudo \
    pandoc \
    pandoc-citeproc \
    pkg-config \
    libnlopt-dev \
    libcurl4-gnutls-dev \
    libcairo2-dev \
    libxt-dev \
    libgsl-dev \
    libssl-dev \
    libssh2-1-dev \
    libssl1.0 \
    libxml2-dev \
    openssl
RUN apt-get update && apt-get install -y \
    libmpfr-dev
RUN apt-get install libcurl4-openssl-dev -y

# install and load ensemblQueryR dependencies
RUN R -e "install.packages('remotes'); \
install.packages('httr'); \
install.packages('xml2'); \
install.packages('dplyr'); \
install.packages('jsonlite'); \
install.packages('purrr'); \
install.packages('tidyr'); \
install.packages('vroom'); \
install.packages('magrittr'); \
install.packages('parallel'); \
library(remotes); \
library(httr); \
library(xml2); \
library(dplyr); \
library(jsonlite); \
library(purrr); \
library(tidyr); \
library(vroom); \
library(magrittr); \
library(parallel)"

# install and load ensemblQueryR (install from clone)
RUN apt-get install -y git
RUN git clone https://github.com/ainefairbrother/ensemblQueryR.git ./ensemblQueryR
RUN R -e "install.packages('devtools'); devtools::install('./ensemblQueryR'); library(ensemblQueryR)"

# clean up after installer
RUN rm -rf /var/lib/apt/lists/*
```

**Figure 3.** Dockerfile containing the setup for the ensemblQueryR Docker image.

'LDpair' and ensemblQueryR's 'ensemblQueryLDwithSNPpair'. Both are configured for single queries and take a pair of variant IDs as input, retrieving the corresponding LD metrics through an API query. To run multiple queries using these single-query functions, both functions were applied over variant identifier (ID) vectors using the base R lapply (Version 4.0.5) [14].

To compare the utility of these functions for LD metric querying in the context of high-throughput workflows, we measured the peak RAM usage and execution speed using the R package peakRAM (version 1.0.2) [18] 10 times for each function and at three query sizes (100, 1,000 and 10,000). The standard deviation (sd) of the performance metric for each function at each time point across the ten iterations was calculated. Mean performance metrics were plotted with error bars to show the sd from the mean



for peak RAM usage (Figure 1a) and time measured in minutes (Figure 1b) for the three query sizes.

### Limitations

It is important to note that there are some limitations to this R package. First, although this package enables high-throughput querying of the Ensembl API, there is an inherent limit to the number of queries that can be submitted arising from the API query limit, which is set at 54,000 requests per hour [5, 6]. On the Ubuntu system used to develop ensemblQueryR (Ubuntu server 16.04 LTS with kernel version 4.4.0-210-generic, total RAM 251 G), 54,000 API requests via ensemblQueryLDwithSNPpairDataframe took ~1.93 h on a single core, making the per-hour request rate 27,867. As such, even query sizes of 54,000 can be run unparallelised and are unlikely to exceed the Ensembl API hourly rate. However, this request limit must be considered by users when applying parallelisation to large queries. The second limitation is that the parallelisation library used to enable the multi-core functionality is the R package ‘parallel’ (version 3.6.2) [19], which works on OSIX systems (Mac, Linux and other Unix-based operating systems) but does not work on Windows.

### Scope for future development

At present, this package provides wrappers for the Ensembl API endpoints that retrieve LD data. However, the Ensembl API offers ~109 other endpoints [5], all of which have the potential to be wrapped into R functions and included in this package. As such, there is scope for the usefulness of this package beyond LD metrics and further development will expand its utility to R users across an array of bioscience disciplines.

## AVAILABILITY OF SOURCE CODE AND REQUIREMENTS

- Project name: ensemblQueryR
- Project home page: https://github.com/ainefairbrother/ensemblQueryR
- Operating system(s): Platform independent
- Programming language: R
- Licence: MIT
- RRID: SCR_024216.

## DATA AVAILABILITY

Snapshots of the code are available in GigaDB [20].

## ABBREVIATIONS

API, Application Programming Interface; eGene, expression of a cis or trans gene; eQTL: Expression Quantitative Trait Loci; GWAS: Genome-Wide Association Study; ID, identifier; LD: Linkage Disequilibrium; RAM: Random Access Memory; REST: Representational State Transfer; rsID: Reference SNP cluster IDs; sd: standard deviation.

## DECLARATIONS

### Ethical approval

The authors declare that ethical approval was not required for this type of research.

## Competing Interests

RHR is currently employed by CoSyne Therapeutics (Lead Bioinformatician). All work performed for this publication was performed in her own time, and not as a part of her duties as an employee.

## Author contributions

AFB and RHR: conceptualisation. AFB: software. AFB and SGR: docker image. AFB: benchmarking. AFB: writing - original draft. AFB, SGR, AH, MR and RHR: writing - review and editing. MR and AH: supervision.

## Funding

AFB was supported through the award of a Biotechnology and Biological Sciences Research Council (BBSRC UK) London Interdisciplinary Doctoral Fellowship. SGR and MR were supported through the award of a Tenure Track Clinician Scientist Fellowship to MR (MR/N008324/1). AH was funded by a BBSRC award (BB/R006075/1).

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
