## [Editor Report]

Editor’s AssessmentENSEMBL is a crucial resource for the vertebrate genomics community. And ensemblQueryR is an R interface to the Ensembl API that facilitates flexible, fast, user-friendly and R workflow integrable querying of the database linkage disequilibrium endpoints. It’s been presented as easy to deploy, and after some debugging the work demonstrate that ensemblQueryR has improved performance memory usage and speed, delivering a 10-fold speed increase over analogous software whilst using a third of the RAM. As the Ensembl API offers >100 other endpoints, all of these have the potential to be wrapped into R functions and included in this package.

---

## [Reviewer Report]

Reviewer name and names of any other individual's who aided in reviewerAziz KhanDo you understand and agree to our policy of having open and named reviews, and having your review included with the published manuscript. (If no, please inform the editor that you cannot review this manuscript.)YesIs the language of sufficient quality?YesPlease add additional comments on language quality to clarify if neededIs there a clear statement of need explaining what problems the software is designed to solve and who the target audience is? YesAdditional CommentsIs the source code available, and has an appropriate Open Source Initiative license <a href="https://opensource.org/licenses" target="_blank">(https://opensource.org/licenses)</a> been assigned to the code?NoAdditional CommentsThere are two license files (the LICENSE, LICENSE.md) and both are conflicting. The authors should use one to make sure it is an appropriate Open Source Initiative license.As Open Source Software are there guidelines on how to contribute, report issues or seek support on the code?NoAdditional CommentsThere are no guidelines on how to contribute.Is the code executable?YesAdditional CommentsIs installation/deployment sufficiently outlined in the paper and documentation, and does it proceed as outlined?YesAdditional CommentsInstallation was straightforward.Is the documentation provided clear and user friendly?YesAdditional CommentsIs there enough clear information in the documentation to install, run and test this tool, including information on where to seek help if required?YesAdditional CommentsIs there a clearly-stated list of dependencies, and is the core functionality of the software documented to a satisfactory level?YesAdditional CommentsThe DESCRIPTION file has all the required packages. Have any claims of performance been sufficiently tested and compared to other commonly-used packages? NoAdditional CommentsAdditional CommentsAre there (ideally real world) examples demonstrating use of the software? YesAdditional CommentsThere are some use cases in the minimal documentation. Some of these are not working. Is automated testing used or are there manual steps described so that the functionality of the software can be verified?YesAdditional CommentsThe package is already in the CRAN, which means it passed its checks. It does have example snippets of code to be run. Any Additional Overall Comments to the AuthorThe ensemblQueryR package offers an R interface for querying Ensembl REST API linkage disequilibrium (LD) endpoints. The package aims to support high-throughput querying, utilizing familiar R object types and parallelization. The package is already in CRAN and comes with Docker/Singularity options. While the paper highlights notable strengths, such as the R interface, high-throughput support, parallelization, and performance improvements, a few points warrant further consideration and clarification.  (1). The package is already in CRAN, which is good, but there are issues with the use-case examples. When I executed the test case to query LD metrics for a variant within a window, the package returned a tibble containing NAs. (2). The title claims the package supports querying "several" Ensembl API endpoints, but the current version is limited to LD endpoints. Revising the title to reflect the package's current capabilities accurately is recommended. (3). Two conflicting license files exist (the LICENSE, LICENSE.md). The authors should select one to ensure it aligns with an appropriate Open Source Initiative license. Additionally, there are no guidelines provided for contributing to the project. (4). I have encountered this R package to get Ensembl data in R using the REST API - https://github.com/dwinter/rensembl. Can the authors comment on this? RecommendationMinor Revisions

---

## [Reviewer Report]

Reviewer name and names of any other individual's who aided in reviewerAndy YatesDo you understand and agree to our policy of having open and named reviews, and having your review included with the published manuscript. (If no, please inform the editor that you cannot review this manuscript.)YesIs the language of sufficient quality?YesPlease add additional comments on language quality to clarify if neededIs there a clear statement of need explaining what problems the software is designed to solve and who the target audience is? YesAdditional CommentsIs the source code available, and has an appropriate Open Source Initiative license <a href="https://opensource.org/licenses" target="_blank">(https://opensource.org/licenses)</a> been assigned to the code?YesAdditional CommentsAs Open Source Software are there guidelines on how to contribute, report issues or seek support on the code?YesAdditional CommentsWhilst nothing explicit is there the code is hosted on GitHub & therefore using the issue tracker would appear to be the right way to do thisIs the code executable?YesAdditional CommentsIs installation/deployment sufficiently outlined in the paper and documentation, and does it proceed as outlined?YesAdditional CommentsIs the documentation provided clear and user friendly?YesAdditional CommentsIs there enough clear information in the documentation to install, run and test this tool, including information on where to seek help if required?YesAdditional CommentsIs there a clearly-stated list of dependencies, and is the core functionality of the software documented to a satisfactory level?YesAdditional CommentsHave any claims of performance been sufficiently tested and compared to other commonly-used packages? YesAdditional CommentsIs test data available, either included with the submission or openly available via cited third party sources (e.g. accession numbers, data DOIs)?NoAdditional CommentsThe code uses data from Ensembl so this is somewhat sufficient and uses rsIDsAre there (ideally real world) examples demonstrating use of the software? YesAdditional CommentsIs automated testing used or are there manual steps described so that the functionality of the software can be verified?YesAdditional CommentsAny Additional Overall Comments to the AuthorThe authors present in their manuscript “ensemblQueryR: fast, flexible and high-throughput querying of Ensembl API endpoints in R”; a library for the popular R programming language and specifically addresses fast retrieval of linkage disequilibrium from Ensembl’s REST APIs. The toolkit provides a convenient set of functions which mediates queries into the Ensembl REST API and presents results in a manner which is optimised and familiar for R users.  Overall I find the manuscript good and provides a useful toolkit for R developers. It is pleasing to see more people use the Ensembl REST APIs and to expand its user base. The analysis benchmarks also appear appropriate. I have only two minor comments.  1. Code examples  Whilst I was able to execute the given example I did encounter some issues. Namely that when executing the code  ensemblQueryR::ensemblQueryLDwithSNPwindow(rsid="rs4129267", r2=0.8, d.prime=0.8, window.size=500, pop="1000GENOMES:phase_3:EUR")  I got the following result  query snp_in_ld r2 d_prime population_name <chr> <lgl> <lgl> <lgl> <lgl> 1 rs4129267 NA NA NA NA  It was disconcerting to not get any results back and now I am unsure if I did the right thing. I then found the GitHub repository was a far better reference for the correct commands to run and would encourage perhaps less information in the manuscript and the authors pointing readers to the GitHub repository or the site https://ainefairbrother.github.io/ensemblQueryR/.  2. Extensions  The authors suggest possible extensions to ensemblQueryR but it would be useful to know if there are any specific developments planned by the authors. RecommendationMinor Revisions